# Are Urologists Ready for Interpretation of Multiparametric MRI Findings? A Prospective Multicentric Evaluation

**DOI:** 10.3390/diagnostics12112656

**Published:** 2022-11-01

**Authors:** Guglielmo Mantica, Nazareno Suardi, Salvatore Smelzo, Francesco Esperto, Francesco Chierigo, Stefano Tappero, Marco Borghesi, Roberto La Rocca, Marco Oderda, Marco Ennas, Armando Stabile, Francesco De Cobelli, Luigi Napolitano, Rocco Papalia, Paolo Gontero, Carlo Introini, Alberto Briganti, Roberto M. Scarpa, Vincenzo Mirone, Francesco Montorsi, Franco Gaboardi, Carlo Terrone, Gianpiero Cardone

**Affiliations:** 1Department of Urology, IRCCS Ospedale Policlinico San Martino, 16132 Genova, Italy; 2Department of Urology, Spedali Civili of Brescia, 25123 Brescia, Italy; 3Department of Urology, San Raffaele Turro Hospital, 20132 Milan, Italy; 4Department of Urology, Campus Biomedico Hospital, University Campus Biomedico, 00128 Rome, Italy; 5Urology Unit, Department of Neurosciences, Reproductive Sciences and Odontostomatology, University of Naples “Federico II”, 80131 Naples, Italy; 6Urology Unit, Department of Surgical Sciences, University of Turin, 10124 Turin, Italy; 7Unit of Urology, Department of Abdominal Surgery, Galliera Hospital, 16128 Genoa, Italy; 8Division of Oncology/Unit of Urology, Urological Research Institute, IRCCS Ospedale San Raffaele, 20132 Milan, Italy; 9Department of Radiology, IRCCS San Raffaele Scientific Institute, Vita-Salute San Raffaele University, 20132 Milan, Italy; 10Department of Surgical and Diagnostic Integrated Sciences (DISC), University of Genova, 16126 Genoa, Italy; 11Diagnostic and Interventional Radiology Department, IRCCS Ospedale San Raffaele-Turro, Università Vita-Salute San Raffaele, 20132 Milan, Italy

**Keywords:** diagnosis, mpMRI, prostate cancer, training

## Abstract

**Aim:** To assess urologists’ proficiency in the interpretation of multiparametric magnetic resonance imaging (mpMRI). **Materials and Methods:** Twelve mpMRIs were shown to 73 urologists from seven Italian institutions. Responders were asked to identify the site of the suspicious nodule (SN) but not to assign a PIRADS score. We set an a priori cut-off of 75% correct identification of SN as a threshold for proficiency in mpMRI reading. Data were analyzed according to urologists’ hierarchy (UH; resident vs. consultant) and previous experience in fusion prostate biopsies (E-fPB, defined as <125 vs. ≥125). Additionally, we tested for differences between non-proficient vs. proficient mpMRI readers. Multivariable logistic regression analyses (MVLRA) tested potential predictors of proficiency in mpMRI reading. **Results:** The median (IQR) number of correct identifications was 8 (6–8). Anterior nodules (number 3, 4 and 6) represented the most likely prone to misinterpretation. Overall, 34 (47%) participants achieved the 75% cut-off. When comparing consultants vs. residents, we found no differences in terms of E-fPB (*p* = 0.9) or in correct identification rates (*p* = 0.6). We recorded higher identification rates in urologists with E-fBP vs. their no E-fBP counterparts (75% vs. 67%, *p* = 0.004). At MVLRA, only E- fPB reached the status of independent predictor of proficiency in mpMRI reading (OR: 3.4, 95% CI 1.2–9.9, *p* = 0.02) after adjusting for UH and type of institution. **Conclusions:** Despite urologists becoming more familiar with interpretation of mpMRI, their results are still far from proficient. E-fPB enhances the proficiency in mpMRI interpretation.

## 1. Introduction

Multiparametric magnetic resonance imaging (mpMRI) of the prostate has become the gold standard imaging modality for suspicious prostate cancer (PCa) [1,2]. Through its different sequences, such as T2, diffusion weighted imaging (DWI), dynamic contrast imaging and spectroscopy, MRI is able to provide information on the cellularity of solid tissues, perfusion parameters in neoplasms and the relative concentration of intracellular metabolites [3].

mpMRI represents a mandatory step in the flowchart of PCa diagnosis as well as a useful tool for staging and preoperative planning of PCa treatment [4,5,6]. The use of mpMRI ranges from the image-fusion technique for the execution of targeted prostate biopsies to an important role in treatment planning, such as the feasibility of nerve-sparing procedures and better assessment of clinical stage [7]. Furthermore, a prebiopsy mpMRI may save unnecessary prostate biopsies and improve the selection of patients for active surveillance in low-risk PCa patients [8].

The interpretation of mpMRI is challenging even in the hands of experienced radiologists as it implies understanding of different MRI sequences, how to manipulate the images and how to assign suspicion scores. Although urologists are not required to have the same skills and precision required as radiologists, the “best practice” requires that the surgeon has adequate skills to read the imaging autonomously and to obtain the basic information for accurate prostate biopsies and treatment delivery [9]. Recently, several training courses have been developed for both radiology residents [10] and urologists [11,12,13], with durable success in increasing diagnostic accuracy. However, to the best of our knowledge, no previous study addressed the ability of urologists (either fully formed or in training) to identify suspicious lesions in mpMRI, with particular attention in identifying those who might be more proficient in the task. To address this void, we relied on a multicentric prospective survey at seven Italian institutions.

## 2. Materials and Methods

Suspicious mpMRI images of 12 patients were selected by an expert uro-radiologist (G.C.), whose interpretation was considered as the gold standard, and shown to urologists from seven institutions (five academic and two non-academic) from August 2019 to November 2020.

### 2.1. Definition of the Test Cases

All the selected scans were attained with a 1.5-T mpMRI study (Achieva and Achieva dStream; Philips Medical Systems, Best, The Netherlands) with a phased-array surface coil and an endorectal coil (BPX-15; Bayer Medical Care, Indianola, PA, USA). All 12 cases presented a PIRADS ≥ 3 lesion [14] (Figure 1 and Figure 2), in details:Case 1: 59 years old, PSA 5.5 ng/mL, no previous biopsies, PIRADS 4, posterior—apical lesion.Case 2: 80 years old, PSA 7.2 ng/mL, no previous biopsies, PIRADS 4, posterior—median lesion.Case 3: 74 years old, PSA 5.4 ng/mL, no previous biopsies, PIRADS 4, anterior—median lesion.Case 4: 77 years old, PSA 5.7 ng/mL, previous negative biopsy, PIRADS 4, anterior—median lesion.Case 5: 73 years old, PSA 5.5 ng/mL, no previous biopsies, PIRADS 5, posterior—apical lesion.Case 6: 59 years old, PSA 5.2 ng/mL, previous negative biopsy, PIRADS 4, anterior—apical lesion.Case 7: 73 years old, PSA 6.5 ng/mL, previous negative biopsy, PIRADS 5, posterior—basal lesion.Case 8: 73 years old, PSA 4.5 ng/mL, previous negative biopsy, PIRADS 4, posterior—median lesion.Case 9: 75 years old, PSA 7.6 ng/mL, no previous biopsies, PIRADS 5, posterior—median lesion.Case 10: 67 years old, PSA 7.9 ng/mL, no previous biopsies, PIRADS 3, posterior—basal lesion.Case 11: 69 years old, PSA 7.8 ng/mL, no previous biopsies, PIRADS 3, anterior—median lesion.Case 12: 67 years old, PSA 5.4 ng/mL, no previous biopsies, PIRADS 4, posterior—median lesion.

### 2.2. Test Administration

The mpMRIs were administered using the RadiAnt DICOM Viewer 4.0.1 software (Medixant, Poznan, Poland). Every participant had a maximum of 10 min to evaluate each mpMRI. Four sequences were shown at the same time in loop: T2 weighted image (T2), diffusion weighted imaging (DWI), apparent diffusion coefficient (ADC), contrast enhancement (CE).

First, each participant answered an anonymous survey (see Appendix A) with demographics and professional data. Second, participants were asked to locate each lesion/nodule on a prostate map (ACR-ESUR-AdMeTech 2019, Pi-Radsv2.1, Page 68, Figure 16: Sector map diagram v2.1). Specifically, the study aimed to only assess the urologists’ ability to find the suspicious nodule and not to assign a PIRADS score, which we consider exclusively a radiologists’ prerogative. Before starting the test, participants were informed about the presence of zero or maximum one lesion/nodule in each mpMRI. Additionally, participants were informed that all patients were biopsy-naïve and harbored PSA levels higher than 4 ng/mL. Five “presenters” (GM, FE, ME, SS, MO) administered the test in the seven different centers. Correct answer was defined as correctly identifying all the following: anterior vs. posterior, left vs. right, apical vs. median vs. base. We relied on an a priori cut-off of 75% correct identification of the site of suspicious lesions as a threshold for proficiency in mpMRI reading (9/12).

### 2.3. Statistical Analysis

Continuous variables were summarized as medians and interquartile ranges (IQR) and categorical variables as the number of subjects and percentage values.

Statistical analyses were based on two steps. First, data were analyzed according to urologists’ hierarchy (resident vs. consultant) as well as according to previous experience in performing fusion prostate biopsies (fPB, defined as <125 vs. ≥125) [15]. Additionally, we tested for differences between non-proficient vs. proficient mpMRI readers. For these analyses, Wilcoxon rank sum test, Pearson’s Chi-squared test and Fisher’s exact test were used where appropriate.

Second, we aimed to assess potential predictors of proficiency in mpMRI reading. For this purpose, we relied on multivariable logistic regression models. Covariates consisted of type of Institution (academic vs. non-academic), previous experience in fPB and urologists’ hierarchy (resident vs. consultant).

For all statistical analyses, R software environment for statistical computing and graphics (version 3.4.3, R Foundation, Vienna, Austria) was used. All tests were 2 sided with a level of significance set at *p* < 0.05.

## 3. Results

Overall, we collected complete data of 73 urologists. Of these, 36 (49%) and 37 (51%) were, respectively, residents and consultants (Table 1A); 54 (74%) vs. 19 participants (26%) worked in academic vs. non-academic institutions, respectively. Median (IQR) age of the participants was 33 (29–50) years. In four centers (43 participants, 59%), oncological multidisciplinary team discussions were part of routine patient care.

Forty-nine participants (67%) were already out of their learning curve for fPB. Similarly, 43 (59%) and 31 (42%) responders came from institutions with more than 300 referrals for PCa and more than 200 prostatectomies per year, respectively.

Overall, 81%, 71%, 38%, 51%, 79%, 42%, 68%, 62%, 77%, 41%, 73%, and 60% of participants correctly identified the lesions from 1 to 12, respectively. When summing up the number of correct responses, 34 (47%) participants achieved the 75% cut-off.

In our cohort of responders, we found no differences in experience in fPB between consultants vs. residents (*p* = 0.9). Similarly, no differences were recorded in the percentage of correctly identified lesions (*p* = 0.6). Only in case number 10, a statistically significant difference was recorded among consultants (59% of correct answers) vs. residents (22%, *p* = 0.001).

In the comparison according to experience in fPB (Table 1B), we recorded a statistically significantly greater percentage of correct responses in urologists outside their learning curve vs. their in-training counterparts (75% vs. 67%, *p* = 0.004). Specifically, the former correctly identified more frequently the lesion of cases 1, 3 and 6 (all *p* < 0.05). 

When analyzing proficient vs. non-proficient readers (Table 1C), we identified statistically significant differences only in experience in fPB (47% vs. 21%, *p* = 0.016). Furthermore, a statistically significantly greater percentage of proficient responders correctly identified the lesions.

In multivariable logistic regression analyses, only experience in fPB reached the status of independent predictor of proficiency in reading mpMRI (OR: 3.4, 95% CI 1.2–9.9, *p* = 0.02) after adjusting for urologists’ hierarchy and type of institution (Table 2).

## 4. Discussion

Since the introduction of mpMRI for PCa diagnosis in clinical practice, the concern of the correct identification of site, size and type of lesion by radiologists, as well as by urologists, has emerged. In fact, although the radiologist is responsible for the diagnosis and attribution of a PIRADS class to prostatic lesions found on mpMRI, it can be postulated that the urologists should be able to correctly identify the lesions in order to perform an accurate prostate biopsy as well as to better plan prostatectomy.

In this light, in recent years, inter-reader agreement in interpreting mpMRI among radiologists has been extensively evaluated [16,17,18,19]. The recent feeling is that in clinical practice, the agreement between radiologists in detecting suspicious lesions on magnetic resonance images could be higher than previously thought [20]. Giganti et al. recently demonstrated a strong reproducibility in the assessment of Prostate Imaging Quality (PI-QUAL) score between radiologists with high expertise in prostate mpMRI [21]. Moreover, an improvement in inter-observer agreement among radiologists in reporting mpMRI was reported after the introduction of PIRADSv2.1 compared with PIRADSv2 [22,23].

Furthermore, some studies evaluated the role of computer-aided diagnosis (CAD) of prostate cancer on mpMRI, using artificial intelligence (AI) in order to reduce missed cancers and unnecessary biopsies and increase inter-observer agreement between clinicians [24]. However, there is insufficient evidence to suggest the clinical deployment of AI algorithms at present.

To date, several studies have tested the proficiency of radiologists in mpMRI interpretation, also addressing the impact of different learning tools advocated to enhance their reading skills [12,25]. Contrarily, only sparse data concerned the ability of urologists in the interpretation of mpMRI images. Kasivisvanathan et al. recently emphasized the promising role of mpMRI teaching courses in improving urologists’ reading ability [11]. However, predictors of higher proficiency in mpMRI interpretation were not provided, nor was detailed information regarding both mpMRI scans and urologists characteristics. We addressed this void relying on a multicentric survey among seven Italian institutions and drew several noteworthy observations.

First, it is worrisome that only 47% of responders achieved our a priori standard of proficiency (75% correct answers). In light of these results, surgeons should still consult radiologists during surgical planning. It is of utmost importance that urological surgeons receive formal training in mpMRI reading.

Second, the least correctly identified lesions were numbers 3, 4, 6 and 10; it is of note that all those nodules were anterior lesions, except for case 10, which was a PIRADS 3 posterior basal lesion. Interestingly, even when responders reached the 75% cut-off of proficiency reading, those lesions were less frequently correctly identified. It can be therefore postulated that anterior and low-PIRADS lesions represent a challenge even for more experienced eyes.

Third, we did not record differences in identifying correct lesions between residents vs. consultants. These results can be justified by the same exposure to fusion prostate biopsy between the two groups (33% vs. 32%, *p* > 0.9). However, it is of note that consultants correctly identified the only PIRADS 3 lesion included in the study (case 10) almost twice as frequently as residents.

Finally, and foremost, experience in fPB seems to play the leading role in achieving proficiency in mpMRI reading. Indeed, when comparing experienced vs. non-experienced, correct responses were recorded in 75% vs. 67%, respectively (*p* = 0.004). Moreover, urologists exposed to fPB were more likely to identify anterior lesions (case 3: 71% vs. 22%, *p* < 0.001; case 6: 71% vs. 29%, *p* < 0.001). Additionally, among those who reached proficiency cut-off, 47% had experience in fPB compared to only 21% of non-proficient readers (*p* = 0.02). This result was further confirmed by MVA analyses, where experience in fPB was the only independent predictor of correct mpMRI reading (OR: 3.4, 95% CI 1.2–9.9, *p* = 0.02) after adjusting for urologists’ hierarchy and type of institution.

Our study is not devoid of limitations; first, both the relatively small number of lesions selected for evaluation and the number of participants could possibly affect the statistical robustness of our analyses. However, we believe that the 12 selected lesions represented a good sample of what urologists can face in their activity and, to the best of our knowledge, our study is the first and the largest to address this literature gap.

Second, our data did not allow us to test for possible predictors of increased difficulty in identification of suspicious nodules as side, site, size of lesions and their PIRADS. Additionally, we only selected mpMRI bearing unifocal suspicious lesions, which does not reflect clinical reality. Indeed, future research should aim at also including in such tests normal mpMRIs as well as those with multiple lesions or benign prostatic diseases (prostatitis, ectopic nodules, microcalcifications, previous biopsies, etc.). Moreover, we feel that radiologists, as well as urologists, should be provided with a short patient summary to help with the correct interpretation of mpMRI sequences.

Third, the study is limited by the lack of a training intervention to see whether the accuracy can be improved with an educational program.

Finally, participants of our study all came from Italian institutions. Despite the effect of including academic vs. non-academic, high vs. low-volume centers and residents vs. consultants, we are not able to conclude that our results are generalizable to other European and non-European settings.

## 5. Conclusions

Our results suggest that experience in fusion prostate biopsies enhances the ability of correctly identify suspicious nodules at mpMRI. Moreover, anterior lesions may be more challenging to locate.

## Figures and Tables

**Figure 1 diagnostics-12-02656-f001:**
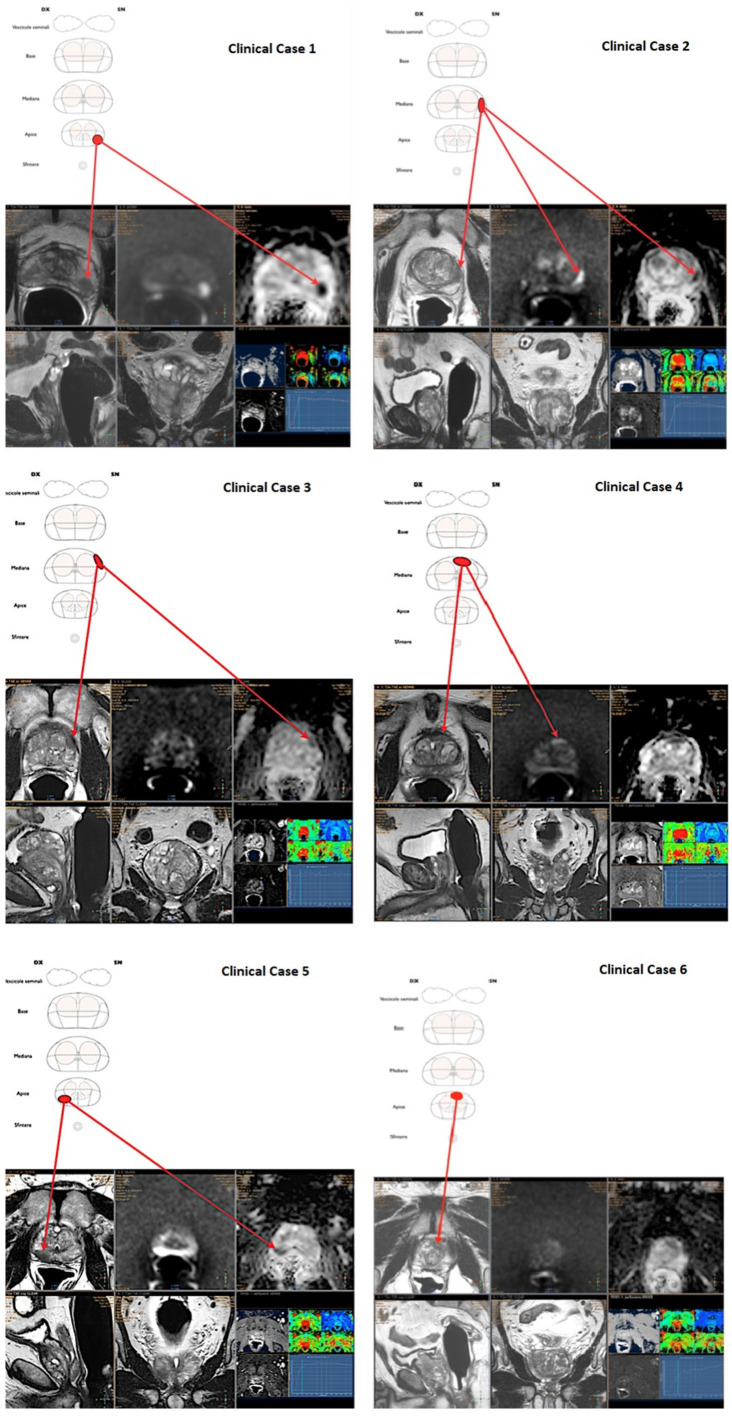
**First set of six** mpMRI images used for urologists’ interpretation skills assessment.

**Figure 2 diagnostics-12-02656-f002:**
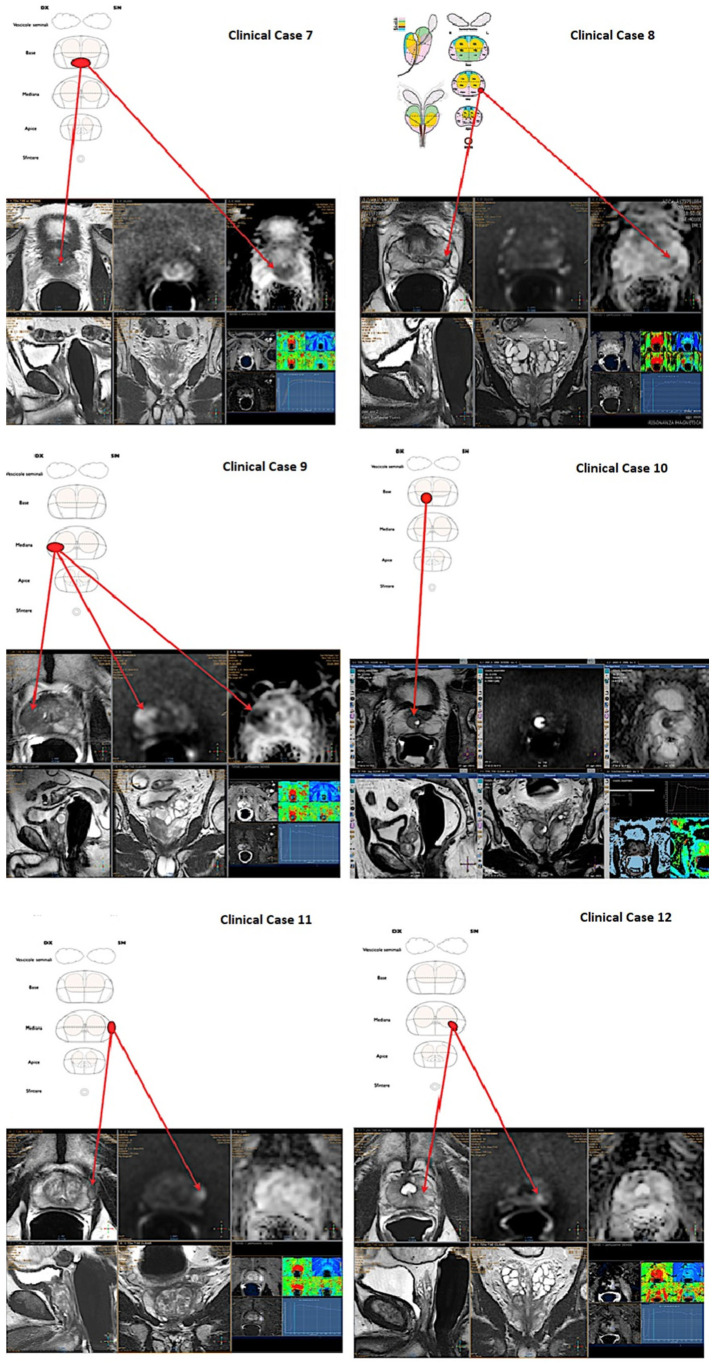
**Second set of six** mpMRI images used for urologists’ interpretation skills assessment.

**Table 1 diagnostics-12-02656-t001:** (**A**–**C**) Characteristics of 72 responders, stratified according to: (**A**) hierarchy (residents vs. consultants); (**B**) experience in fusion prostate biopsy (non-experienced vs. experienced); (**C**) proficiency in mpMRI reading (≥75% of correct identifications; non-proficient vs. proficient).

	(A) Comparison According to Hierarchy			(B) Comparison According to Experience in fPB			(C) Comparison According to Proficiency in mpMRI Reading (≥75% of Correct Identifications)		
	Residents, n = 36 (49%) ^1^	Consultants, n = 37 (51%) ^1^	*p*-Value ^2^	Non- Experienced, n = 49 (67%) ^1^	Experienced, n = 24 (33%) ^1^	*p*-Value ^2^	Non-Proficient, n = 39 (53%) ^1^	Proficient, n = 34 (47%) ^1^	*p*-Value ^2^
**Age**	28 (27, 30)	50 (39, 58)	<0.001	35 (29, 52)	32 (29, 40)	0.6	31 (28, 50)	34 (29, 50)	0.8
**Number of PCa diagnosed yearly (per center)**	300 (250, 500)	300 (250, 500)	0.5	300 (250, 500)	300 (250, 500)	0.5	500 (275, 500)	250 (250, 500)	0.021
**Number of RP performed yearly (per center)**	130 (100, 300)	130 (100, 300)	0.6	130 (100, 300)	130 (120, 250)	0.6	250 (125, 300)	120 (100, 300)	0.058
**Number of correct identifications**	8.0 (6.8, 9.0)	8.0 (6.0, 10.0)	0.6	8.0 (5.0, 9.0)	9.0 (8.0, 10.0)	0.004	8.0 (5.0, 9.0)	9.0 (8.0, 10.0)	0.004
**Percentage of correct identifications**	67 (56, 75)	67 (50, 83)	0.6	67 (42, 75)	75 (67, 83)	0.004	67 (42, 75)	75 (67, 83)	0.004
**Institution**			<0.001			0.5			0.5
University hospital	34 (94%)	20 (54%)		35 (71%)	19 (79%)		35 (71%)	19 (79%)	
Non university hospital	2 (5.6%)	17 (46%)		14 (29%)	5 (21%)		14 (29%)	5 (21%)	
**Experience in Prostate Biopsy**			0.2			0.003			0.13
No	9 (25%)	5 (14%)		14 (29%)	0 (0%)		10 (26%)	4 (12%)	
Yes	27 (75%)	32 (86%)		35 (71%)	24 (100%)		29 (74%)	30 (88%)	
**Experience in Fusion Prostate Biopsy**			>0.9			-			0.016
No	24 (67%)	25 (68%)		-	-		31 (79%)	18 (53%)	
Yes	12 (33%)	12 (32%)		-	-		8 (21%)	16 (47%)	
**Involvement in diagnosis and management of PCa**			0.4			0.2			0.027
No	4 (11%)	2 (5.4%)		6 (12%)	0 (0%)		6 (15%)	0 (0%)	
Yes	32 (89%)	35 (95%)		43 (88%)	24 (100%)		33 (85%)	34 (100%)	
**Prostate cancer multidisciplinary team present**			0.023			0.050			0.6
No	10 (28%)	20 (54%)		24 (49%)	6 (25%)		17 (44%)	13 (38%)	
Yes	26 (72%)	17 (46%)		25 (51%)	18 (75%)		22 (56%)	21 (62%)	
**Clinical case 1**			0.2			0.027			0.007
Incorrect	9 (25%)	5 (14%)		13 (27%)	1 (4.2%)		12 (31%)	2 (5.9%)	
Correct	27 (75%)	32 (86%)		36 (73%)	23 (96%)		27 (69%)	32 (94%)	
**Clinical case 2**			0.9			0.11			0.003
Incorrect	10 (28%)	11 (30%)		17 (35%)	4 (17%)		17 (44%)	4 (12%)	
Correct	26 (72%)	26 (70%)		32 (65%)	20 (83%)		22 (56%)	30 (88%)	
**Clinical case 3**			0.6			<0.001			0.017
Incorrect	21 (58%)	24 (65%)		38 (78%)	7 (29%)		29 (74%)	16 (47%)	
Correct	15 (42%)	13 (35%)		11 (22%)	17 (71%)		10 (26%)	18 (53%)	
**Clinical case 4**			>0.9			0.2			<0.001
Incorrect	18 (50%)	18 (49%)		27 (55%)	9 (38%)		30 (77%)	6 (18%)	
Correct	18 (50%)	19 (51%)		22 (45%)	15 (62%)		9 (23%)	28 (82%)	
**Clinical case 5**			0.4			0.12			<0.001
Incorrect	6 (17%)	9 (24%)		13 (27%)	2 (8.3%)		14 (36%)	1 (2.9%)	
Correct	30 (83%)	28 (76%)		36 (73%)	22 (92%)		25 (64%)	33 (97%)	
**Clinical case 6**			0.9			<0.001			0.008
Incorrect	21 (58%)	21 (57%)		35 (71%)	7 (29%)		28 (72%)	14 (41%)	
Correct	15 (42%)	16 (43%)		14 (29%)	17 (71%)		11 (28%)	20 (59%)	
**Clinical case 7**			0.4			0.4			<0.001
Incorrect	13 (36%)	10 (27%)		17 (35%)	6 (25%)		20 (51%)	3 (8.8%)	
Correct	23 (64%)	27 (73%)		32 (65%)	18 (75%)		19 (49%)	31 (91%)	
**Clinical case 8**			0.6			0.7			0.004
Incorrect	15 (42%)	13 (35%)		18 (37%)	10 (42%)		21 (54%)	7 (21%)	
Correct	21 (58%)	24 (65%)		31 (63%)	14 (58%)		18 (46%)	27 (79%)	
**Clinical case 9**			0.4			0.13			0.001
Incorrect	7 (19%)	10 (27%)		14 (29%)	3 (12%)		15 (38%)	2 (5.9%)	
Correct	29 (81%)	27 (73%)		35 (71%)	21 (88%)		24 (62%)	32 (94%)	
**Clinical case 10**			0.001			>0.9			<0.001
Incorrect	28 (78%)	15 (41%)		29 (59%)	14 (58%)		30 (77%)	13 (38%)	
Correct	8 (22%)	22 (59%)		20 (41%)	10 (42%)		9 (23%)	21 (62%)	
**Clinical case 11**			0.6			0.2			<0.001
Incorrect	11 (31%)	9 (24%)		16 (33%)	4 (17%)		19 (49%)	1 (2.9%)	
Correct	25 (69%)	28 (76%)		33 (67%)	20 (83%)		20 (51%)	33 (97%)	
**Clinical case 12**			0.3			0.4			<0.001
Incorrect	12 (33%)	17 (46%)		21 (43%)	8 (33%)		24 (62%)	5 (15%)	
Correct	24 (67%)	20 (54%)		28 (57%)	16 (67%)		15 (38%)	29 (85%)	

^1^ n (%); ^2^ Wilcoxon rank sum test, Pearson’s Chi-squared test and Fisher’s exact test.

**Table 2 diagnostics-12-02656-t002:** Multivariable logistic regression model testing possible predictors of proficiency in mpMRI reading.

	OR ^1^	95% CI ^1^	*p*-Value
**Role (ref. Resident)**	1.38	0.47, 4.20	0.5
**Institution (ref. University hospital)**	0.73	0.20, 2.50	0.6
**Experience in fPB (ref. No)**	3.40	1.24, 9.94	0.020

^1^ OR = odds ratio, CI = confidence interval.

## Data Availability

The data and the code used for the analyses will be made available upon request.

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
