# Peer review of "Are Urologists Ready for Interpretation of Multiparametric MRI Findings? A Prospective Multicentric Evaluation"

_diagnostics, 2022, doi:10.3390/diagnostics12112656_

Round 1
Reviewer 1 Report
Congratulations for this interesting paper evaluating Urologists ability in interpretation of mpMRI and the need for improving this ability by taking part in MRI courses and including them in urology training in order to improve standards of our patient Management.
Author Response
We thank the reviewer for the comment. We really appreciate your contribution.
Reviewer 2 Report
Mantica et al., designed a test to assess urologists’ proficiency in the interpretation of mpMRI in this manuscript. The test demonstrated that urologists are becoming familiar with interpretation of mpMRI. E-fPB enhances the proficiency in mpMRI interpretation. However, this study showed some interesting results, these results are not well presented and lack of comparative analysis with others. There are many reports and experiences related to the interpretation mpMRI, including training, an online ibook. The authors did not clearly justify the importance of this designed test, which greatly impaired the reviewer’s enthusiasm to give the positive comments on this work. In addition, some parts of this manuscript should be rephrased and reorganized to make it sound reasonable.
Author Response
We thank the reviewer for the insightful comment. The following sentences were added in the Discussion section in order to strengthen the message of the manuscript.
Discussion, page 15, line 238: “To date, several studies tested the proficiency of radiologists in mpMRI interpretation, also addressing the impact of different learning tools advocated to enhance their reading skills. 12,25 Contrarily, only sparse data concerned the ability of urologists in the interpretation of mpMRI images. Kasivisvanathan et al. recently emphasized the promising role of mpMRI teaching courses in improving urologists’ reading ability. 11 However, predictors of higher proficiency in mpMRI interpretation were not provided, as well as detailed information regarding both mpMRI scans and urologists characteristics. We addressed this void relying on a multicentric survey among seven Italian Institutions and drew several noteworthy observations”.
References:
- Kasivisvanathan V, Ambrosi A, Giganti F, Chau E, Kirkham A, Punwani S, Allen C, Emberton M, Moore CM. A Dedicated Prostate MRI Teaching Course Improves the Ability of the Urologist to Interpret Clinically Significant Prostate Cancer on Multiparametric MRI. Eur Urol. 2019 Jan;75(1):203-204. doi: 10.1016/j.eururo.2018.09.033. Epub 2018 Oct 14. PMID: 30327275.
- Christidis D, McGrath S, Leaney B, O'Sullivan R, Lawrentschuk N. Interpreting Prostate Multiparametric Magnetic Resonance Imaging: Urologists' Guide Including Prostate Imaging Reporting and Data System. Urology. 2018 Jan; 111:136-138. doi: 10.1016/j.urology.2017.08.013. Epub 2017 Aug 18. PMID: 28823635.
- Akin O, Riedl C, Ishill N, Moskowitz C, Zhang J, Hricak H. Interactive dedicated training curriculum improves accuracy in the interpretation of MR imaging of prostate cancer. Eur Radiol.2010 Apr;20(4):995-1002. doi: 10.1007/s00330-009-1625-x.
We hope that the reviewer will consider the above modifications satisfactory.
Reviewer 3 Report
Dear Authors,
In this study 12 multiparametric MRI were analyzed by 73 urologist from 7 Italian institutions. The study goal was to evaluate the proficiency of urologists in interpretation of multiparametric MRI of prostate’s lesions. The authors showed ability of urologist to interprets accurately multiparametric MRI and the interest to get experience in fusion prostate biopsies.
However, there are a number of issues to be addressed:
1. Abstract – p1, l34. Please define mpMRI.
2. Abstract – p2,l38. Please define UH.
3. Materials and methods – As the quality of the image depends on the type of MR scanner, coils used and the sequence applied it will be interesting to have this information in the method sections. Also, same MR scanner and same sequences were applied for each patient.
4. Results – Where begin the results section?
5. Results – Table 1. The table 1 is quite confusing and very long (9 pages!). Is it possible to separate the table to provide a more readable version?
6. Results. The numbers of cases are quite low to investigate the reasons why lesions were least correctly identified. As the authors suggest, it can be due to the location of the lesion or a low Pi-RADS. Also, image quality can have an impact. A larger cohort is required to evaluate the impact of lesions types in the ability to correctly identified the lesions.
Author Response
Reviewer #3
Dear Authors,
In this study 12 multiparametric MRI were analyzed by 73 urologists from 7 Italian institutions. The study goal was to evaluate the proficiency of urologists in interpretation of multiparametric MRI of prostate’s lesions. The authors showed ability of urologist to interprets accurately multiparametric MRI and the interest to get experience in fusion prostate biopsies.
However, there are a number of issues to be addressed:
- Abstract – p1, l34. Please define mpMRI.
Response: We thank the reviewer for the comment. We defined mpMRI as recommended.
Abstract, page 4, line 73: “multiparametric Magnetic Resonance Imaging (mpMRI)”.
- Abstract – p2, l38. Please define UH.
Response: We thank the reviewer for the comment. The definition of UH can be found in the materials and methods section of the abstract (Abstract, page 4, line 79).
- Materials and methods – As the quality of the image depends on the type of MR scanner, coils used and the sequence applied it will be interesting to have this information in the method sections. Also, same MR scanner and same sequences were applied for each patient.
Response: We thank the reviewer for the comment. We agree with the reviewer. The following sentence was added:
Materials and methods, page 6, line 125: “All the selected scans were attained with a 1.5-T mpMRI study (Achieva and Achieva dStream; Philips Medical Systems, Best, The Netherlands) with a phased-array surface coil and an endorectal coil (BPX-15; Bayer Medical Care, Indianola, PA, USA)”.
We hope that the reviewer will consider the above modifications satisfactory.
- Results – Where begin the results section?
Response: The Results section begins at line 173, page 8.
- Results – Table 1. The table 1 is quite confusing and very long (9 pages!). Is it possible to separate the table to provide a more readable version?
Response: We thank the reviewer for the comment. We agree with the reviewer. Based on the reviewer’s suggestion we consistently restructured the entire Table 1.
We hope that the reviewer will consider the applied modifications satisfactory.
- The numbers of cases are quite low to investigate the reasons why lesions were least correctly identified. As the authors suggest, it can be due to the location of the lesion or a low Pi-RADS. Also, image quality can have an impact. A larger cohort is required to evaluate the impact of lesions types in the ability to correctly identified the lesions.
Response: We thank the reviewer for the comment. Unfortunately, a larger cohort is not currently available. We stressed the concept of limited sample size of MRI scans as well as MRI readers in the limitations section ([..] first, both the relatively small number of lesions selected for evaluation and the number of participants could possibly affect the statistical robustness of our analyses. However, we believe that the 12 selected lesions represented a good sample of what urologists can face in their activity and, to the best of our knowledge, our study is the first and the largest to address this literature gap”).
We hope that the reviewer will consider this explanation satisfactory.
Round 2
Reviewer 2 Report
I am fine with authors' revision. No more comments from me.
Author Response
We would like to thank Reviewer 2 for the comment.
Reviewer 3 Report
I thank the authors for the changes they have been made.
Author Response
We would like to thank Reviewer 3 for the comment.